# Pathogenesis of Hepatocellular Carcinoma: The Interplay of Apoptosis and Autophagy

**DOI:** 10.3390/biomedicines11041166

**Published:** 2023-04-13

**Authors:** Elias Kouroumalis, Ioannis Tsomidis, Argyro Voumvouraki

**Affiliations:** 1Department of Gastroenterology, PAGNI University Hospital, University of Crete School of Medicine, 71500 Heraklion, Crete, Greece; 2Laboratory of Gastroenterology and Hepatology, University of Crete Medical School, 71500 Heraklion, Crete, Greece; 31st Department of Internal Medicine, AHEPA University Hospital, 54621 Thessaloniki, Central Macedonia, Greece

**Keywords:** hepatocellular carcinoma, disease associations, autophagy, apoptosis, ferroptosis

## Abstract

The pathogenesis of hepatocellular carcinoma (HCC) is a multifactorial process that has not yet been fully investigated. Autophagy and apoptosis are two important cellular pathways that are critical for cell survival or death. The balance between apoptosis and autophagy regulates liver cell turnover and maintains intracellular homeostasis. However, the balance is often dysregulated in many cancers, including HCC. Autophagy and apoptosis pathways may be either independent or parallel or one may influence the other. Autophagy may either inhibit or promote apoptosis, thus regulating the fate of the liver cancer cells. In this review, a concise overview of the pathogenesis of HCC is presented, with emphasis on new developments, including the role of endoplasmic reticulum stress, the implication of microRNAs and the role of gut microbiota. The characteristics of HCC associated with a specific liver disease are also described and a brief description of autophagy and apoptosis is provided. The role of autophagy and apoptosis in the initiation, progress and metastatic potential is reviewed and the experimental evidence indicating an interplay between the two is extensively analyzed. The role of ferroptosis, a recently described specific pathway of regulated cell death, is presented. Finally, the potential therapeutic implications of autophagy and apoptosis in drug resistance are examined.

## 1. Introduction

Hepatocellular carcinoma (HCC) is a very complex world health problem. Approximately 905,677 new cases and 830,180 HCC-related deaths were reported in 2020 [1]. The estimation of more than 1 million deaths caused by HCC by 2030 has been predicted [2]. HCC-associated liver diseases are chronic viral hepatitis B (HBV) and hepatitis C virus (HCV) infection [3], non-alcoholic steatohepatitis (NASH) [4] and alcoholic liver disease (ALD). The etiological risk for HCC varies according to the geographical location [5]. Thus, in the United States, 54.9%, 16.4%, 14.1% and 9.5% of HCC cases are associated with the four commonest liver diseases HCV, HBV, NAFLD/NASH and ALD [6]. In Asia, NAFLD-associated HCC is lower compared to the West [7]. However, the etiology of HCC has changed over the last 20 years with a progressive increase in non-viral cases, such as metabolic HCC and a concomitant decline in viral etiology [8]. This was true in a study from Crete, where the initial high hepatitis C virus association decreased, and alcohol ranked first among risk factors for HCC. Non-alcoholic fatty liver disease was also continually increased as an important risk of HCC [9].

The development of HCC is associated with some form of cellular death, which may be either programmed (PCD) (such as apoptosis, necroptosis and autophagy-dependent cell death) or non-programmable (such as pyroptosis and necrosis) [10,11,12,13]. Apoptosis is probably the commonest cause of PCD. Characteristically, apoptosis does not elicit inflammation because apoptotic bodies are engulfed by macrophages and are degraded by lysosomes in autophagy [14,15]. Autophagy is an important degradation process of cellular contents, leading to the recirculation of structural components of the cell and improved survival. Autophagy-dependent cell death is a rare kind of PCD [16,17,18]. Lysosomes are the most important subcellular organelles involved in the autophagic degradation of protein aggregates [19,20]. Ferroptosis is a different form of PCD as it depends on excessive iron and lipid peroxidation [21,22] and is closely related to a specific form of autophagy called ferritinophagy, which is a critical part of the turnover of cellular iron through the autophagic degradation of ferritin [23]. In addition, other forms of autophagy, such as lipophagy, and heat shock protein 90 (HSP90)-mediated chaperone-mediated autophagy (CMA) may induce ferroptosis by promoting lipid peroxidation. The purpose of this review is to present the current views on HCC pathogenesis and the complex interplay of autophagy, apoptosis and ferroptosis in the pathophysiology and treatment of HCC.

## 2. Pathogenesis of HCC

The cells of origin of HCC are not clear. Experimental evidence supports the implication of transformed mature hepatocytes as the cell of origin, but also the possibility that the source is liver stem cells [24,25].

HCC is the result of either mutations, such as those in the TERT promoter or p53 suppressor gene [26,27], or epigenetic modifications. Some of them are directly involved or activate important signaling pathways leading to HCC [28]. Genes coding for several signaling pathways, such as Wnt/β-catenin, oxidative stress, AKT/mTOR and MAP kinase, are often mutated in HCC and are used for the molecular classification of HCC [29]. All these molecular abnormalities are triggered by external factors, such as alcohol consumption, viral infection and abnormal nutrition. In general, three mechanisms are implicated in the initiation and progress of HCC, namely, persistent liver inflammation, endoplasmic reticulum (ER) stress and abnormalities of cell signaling pathways [5]. Inflammation is an important pathogenetic factor irrespective of the etiology of the liver disease that leads to HCC [30]. Inflammation starts when hepatocytes undergoing programmed or accidental death liberate factors, such as HMGB1 and HDGF, to initiate an inflammatory response [31]. Different inflammasomes, particularly the nucleotide-binding oligomerization domain, leucine-rich repeat and pyrin domain containing 3 (NLRP3), are activated and lead to the release of the pro-inflammatory cytokine IL-1β. NLRP3 activation is mostly triggered by the production of ATP from the mitochondria of the damaged cells [32] and lysosomal disruption [33]. NLRP3 inflammasome activation in hepatocytes is a two-step process. Priming is the first step when damage-associated molecular patterns (DAMPs) from damaged cells and pathogen-associated molecular patterns (PAMPs) stimulate TLR receptors, followed by translocation of NF-kB to the nucleus and increase in pro-IL-1β and pro-IL-18 expression. The second step is triggered when extracellular ATP or active lysosomal enzymes finally lead to the activation of caspase-1. Cleavage by caspase-1 turns pro-cytokines into mature IL-1β and IL-18, while the cleavage of gasdermin D leads to a programmed cell death called pyroptosis. In this canonical activation of pyroptosis, fragments of gasdermin D form pores in the plasma membrane, killing the cell and releasing IL-1β and IL-18, which aggravates inflammation [34,35].

In the non-canonical pyroptosis, lipopolysaccharides (LPSs), from Gram-negative bacteria, turn the pro-caspases 4, 5 and 11 into active enzymes that cleave gasdermin D. The pores in the plasma membrane are formed, but without maturation, and release IL-1β and IL-18. This is not always the case, as caspase-11 may activate the NLPR3-dependent caspase-1 inflammasome and indirectly stimulate the release of intracellular cytokines [36,37].

Interestingly, inflammasome-mediated pyroptosis also occurs in non-parenchymal liver cells, implicating the gut microbiota. DAMPS and gut-derived PAMPs activate Kupffer cells that produce IL-1β and TNF. NLRP3 activation in hepatic stellate cells (HSCs) promotes the production of the profibrogenic cytokine and induces the expression of the profibrogenic molecule TGF-β. These events in concert lead to liver inflammation and fibrosis, being the link between liver damage and hepatocellular carcinoma [38,39].

It was suggested that cathepsin B from lysosomes is the triggering stimulus of the NLRP3 inflammasome, which is mostly mediated by the release of Cathepsin B [40,41,42]. However, mice macrophages deficient in cathepsin B showed comparable NLRP3 inflammasome activation with wild-type animals [43], suggesting that other products contribute to NLRP3 inflammasome activation. Cathepsins B, L, C, S and X are probable candidates in NLRP3 inflammasome activation by silica particles [44]. In an adenovirus infection, cathepsin B release is also a mediator of inflammation, but reactive oxygen species (ROS) inhibition reduces IL-1β secretion, indicating that ROS production might be the mechanism of the induction of inflammasome activation by cathepsin B [45].

### 2.1. Endoplasmic Reticulum (ER) and Oxidative Stress

ER stress is caused by the accumulation of unfolded or misfolded proteins in the ER lumen. Pathogens, mutations and an increased metabolic rate led to an increase in the protein secretory load of the hepatocyte. The accurate monitoring of protein folding is not maintained in ER, inducing the unfolding protein response (UPR) either to normalize protein synthesis or to induce cell death in severe ER stress [46,47]. ER stress in murine hepatocytes activates inflammatory pathways, such as NF-kB and TNF, leading to HCC induction [48]. Chronic ER stress and increased UPR activity have been implicated in the development of HCC and are present in HCC tumors irrespective of grade or stage [49,50,51,52].

### 2.2. Abnormalities of Signaling Pathways


*mTOR pathway*


The abnormal activation of the oncogenic phosphoinositide 3-kinase/protein kinase B/mammalian target of rapamycin (PI3K/AKT/mTOR) signaling is associated with HCC. This is not unexpected, as this pathway is involved in various cellular functions, such as cellular proliferation, differentiation, apoptosis and metabolism [53]. The AKT/mTOR pathway is known to interfere with aerobic glycolysis regulating three limiting enzymes in the glycolytic pathway (hexokinase 2, phosphofructokinase 1, and pyruvate kinases type M2), a fact that is crucial for HCC progression [54,55,56,57]. The mTOR pathway is analyzed later, as it is implicated in the interplay between autophagy and apoptosis.


*Wnt/β-catenin pathway*


The deregulation of Wnt/β-catenin signaling is critical in human HCC [58,59]. A total of 35% of human HCC tumors had a gain-of-function mutation of CTNNB1 encoding β-catenin and loss-of-function mutation of AXIN1 [60,61]. Resistance to sorafenib and regorafenib treatment was attributed to activated Wnt/β-catenin in HCC patients [62,63]. There is evidence that a second signal is required because Wnt/β-catenin alone is not sufficient to induce hepatocarcinogenesis. Oncogenic mutations of β-catenin cooperate with other oncogenes, such as c-Met [64,65,66] and K-RasV12 [67]. After activation, β-catenin induces several downstream targets that are implicated in HCC induction. c-MYC is one of the best-studied down-stream effectors of β-catenin [68,69].


*miRNAs*


miRNAs are regulators of several tumor-related genes in carcinogenesis, acting either as oncogenes or tumor suppressor genes. miRNAs are classified according to their implication in the main molecular pathways leading to HCC tumorigenesis [70]. miR-30a, miR-365, miR-526a, miR-377, miR-199a-5p and miR-330 all were implicated in apoptosis regulation and were either upregulated or downregulated in HCC [71]. Other mRNAs are involved in the repression of the PI3K/AKT/mTOR pathway or the Wnt/β-catenin pathway [53]. Tumor suppressor miRNAs are associated with either HCC initiation and progression [72] or with metastasis and recurrence [73]. Similarly, some pro-oncogenic miRNAs are associated with HCC initiation and progression [74], or are involved in HCC recurrence and metastasis [75].

### 2.3. Additional Factors Are Involved in HCC Pathogenesis


*Exosomes*


Exosomes transfer proteins, DNAs and RNAs, such as miRNAs, long non-coding RNAs (lncRNAs) and messenger RNAs (mRNAs), between HCC and normal cells. Exosomes initiate either local or systemic reactions, participating to the initiation and progression of HCC. Exosomes are used as biomarkers and therapeutic tools in HCC [76,77].


*Ferroptosis*


Ferroptosis is an iron-dependent process of regulated cell death, with accumulation of lipid peroxides that causes damage to liver cells and is associated with the development of HCC. It is analyzed later in the paper.


*Microbiota*


Microbial products in the bowel are deeply involved in HCC pathogenesis. Bacterial products from the gut microbiota can directly or indirectly damage DNA through the pro-duction of ROS [78,79]. The altered composition of the gut microbiota may be one of the mechanisms underlying the action of aflatoxin and other mycotoxins as powerful inducers of HCC [80]. In addition to gut microbial products, several studies have incriminated specific gut microbiota in association with HCC. Enterococcus faecalis is increased in HCV-induced HCC development [81]. A similar study showed a significant increase in Enterococcus species in patients with viral and alcoholic cirrhosis leading to HCC [82]. Microbiotas also interfere with bile acid metabolism, contributing to HCC development. Bile acid metabolites, produced by the gut microbiota, can cause inflammation, ROS overproduction and a reduction in apoptosis in the liver, finally leading to the development of HCC. Moreover, they can modulate the function of liver immune cells, affecting HCC progression. They can also indirectly contribute to the activation of the TLR4 receptor in hepatocytes and Kupffer cells. Bile acids increase gut permeability, acting on the tight junctions, and allow for an increased transportation of LPS to the liver, thus promoting angiogenesis and the downregulation of tumor suppressor miRNAs [83].


*Calcium*


Ca^2+^ is present in various cell compartments transported among them by transporters and exchangers, collectively known as the Ca^2+^ transportome. The impairment of the Ca^2+^ transportome contributes to HCC initiation, the formation of metastatic cells and reduction in cell death [84].


*Autophagy and Apoptosis*


These two critical parameters in the initiation and progress of HCC are analyzed separately. Two very informative reviews on the pathogenesis of HCC have been recently published [5,38].

Detailed pathophysiological factors implicated in HCC pathogenesis, including gene mutation and epigenetic changes, have been recently reviewed [85].

## 3. HCC Related to Specific Diseases

The pathogenesis of HCC has some discrete characteristics associated with the etiology of the liver disease.

### 3.1. HBV

HBV, as many other oncogenic viruses, does not directly lead to the development of cancer. It is the interaction with host factors that first initiates pre-neoplasia and then carcinoma [3]. Chronic inflammation from virally induced immune reactions is due to inflammasome activation, increased secretion of pro-inflammatory cytokines and increased levels of ROS within the liver microenvironment, which finally determines the development of cancer [86]. A chronic HBV infection leads to long-lasting hepatic inflammation, inducing cirrhosis and HCC progression due to increased hepatocyte turnover rates and the accumulation of mutations [87]. The alterations of platelets could also play a role in hepatocarcinogenesis [88,89].

In addition to the critical role of inflammation, HBV directly affects hepatocarcinogenesis, unlike HCV. This is because HBV integrates its genome into the DNA of the host, leading to genomic instability and mutagenesis in both proto-oncogenes and tumor suppressor genes [90]. HBV integration alters the tumor suppressor gene p53 or the combined p53–Rb pathway. Overall, most HCC cases harbor mutations in the component genes of either the p53 or the Rb pathway alone or of the combined p53–Rb pathway [91]. These alterations are associated with the inhibition of apoptosis [92]. This is the main mechanism through which HBV induces HCC in the absence of cirrhosis. In the presence of cirrhosis, hepatocarcinogenesis is multifactorial as there are several target genes impaired by HBV genome integration [93,94,95]. The integration of HBV DNA is constantly detected in 80% to 90% of tumor tissues and in 30% of non-HCC tissues next to HCC [93], even prior to the induction of HCC [96]. It is conceivable, therefore, that a hidden HBV infection may exist in HBsAg-negative patients and induce hepatocarcinogenesis [97]. The increased expression of truncated HBsAg, HBcAg and HBx proteins favors HCC development through the endoplasmic reticulum and mitochondrial stress [98,99].

#### 3.1.1. The Important Role of HBx

HBx plays its role through several mechanisms [100,101]. HBx expression might induce hepatocarcinogenesis by interfering with telomerase activity during hepatocyte proliferation [96,102], upregulating the activation of human TERT [103,104,105]. In addition, the X protein interacts with several nuclear transcription factors and signal transduction pathways [106,107]. Among the most important deregulated pathways are the Wnt/β-catenin, the PI3K/Akt/mTOR and the Ras/Raf/mitogen-activated protein kinases (MAPK) pathways [108]. HBx and pre-S proteins activate mTOR signaling during an HBV infection and increase cell proliferation and angiogenesis [109,110]. Moreover, HBx has either anti-apoptotic [111,112,113] or pro-apoptotic activity [114]. These effects collectively lead to uncontrolled malignant transformation.

Epigenetic changes refer to chromatin changes without interference with the DNA sequence and include DNA methylation, histone modification and RNA-related silencing [115]. HBx causes epigenetic hyper- or hypo-methylation of the DNA and the tumor suppressor genes, inducing chromosomal instability [116,117]. HBx also promotes the acetylation of the histones H3 and H4, contributing to the pathogenesis of HCC [118,119,120,121].

#### 3.1.2. The Role of RNAs

Long non-coding RNAs (lncRNAs) and circular RNAs (circRNAs) also contribute to the initiation and progression of HBV-related HCC [122,123,124]. Several microRNAs can be regulated by HBV infection and promote hepatocarcinogenesis [125,126]. They modulate the Wnt/β-catenin signaling pathway, leading to the development of HCC [127].

The role of superinfection with HDV in the development of HCC is not clear. A direct oncogenic effect of HDV has not been unequivocally demonstrated [128]. A molecular signature of HDV–HCC different from HBV–HCC in malignant and non-malignant hepatocytes has been reported [129]. The pathogenesis of HBV-associated HCC has been recently reviewed [100,130].

### 3.2. HCV

The pathogenesis of HCC in HCV has certain similarities and differences compared to that in HBV. Thus, DNA damage also occurs during HCV replication, causing genomic instability and leading to hepatocarcinogenesis [131]. The role of chronic inflammation is as important in HCV as it is in HBV in relation to carcinogenesis. However, it has certain discrete characteristics, as inflammation is used for the immune escape of the virus. Macrophages are activated by the core and NS3 proteins of HCV, triggering the NLRP3 inflammasome and inducing the secretion of pro-inflammatory cytokines. IL-18 secretion induces NK cell activation. In addition, IL-1β and IL-6 production by macrophages support the activation of HSCs, increasing collagen deposition and fibrosis. IL-1β, IL-6 and TNF-α secretion may lead to malignant induction [132,133]. The chronic inflammatory environment in combination with certain viral proteins leads to a continuous activation of signaling pathways associated with hepatocyte survival, such as STAT3 and NF-kB. STAT3 is also involved in the development of myeloid-derived suppressor cells (MDSCs), which produce IL-10 favoring the expansion of regulatory T cells. Tregs impair immune response, and this is further accentuated by the increased expression of programmed cell death protein 1 (PD-1) and Fas ligand (FasL), driving the HCV-specific cytotoxic T lymphocytes to their apoptosis [134]. Epigenetic abnormalities similar to HBV also favor HCV-induced hepatocarcinogenesis. Thus, DNA methylation, histone modifications and microRNAs are also involved in the development of HCC [135]. LncRNAs and microRNAs similarly contribute to the induction and progression of HCV-associated HCC. miR-373 forms a complex with LINC00657, promoting uncontrolled cell growth [136]. The hypermethylation of certain promoter regions suppress mRNA expression, favoring the progression of HCV-associated HCC [137].

The major difference between HBV- and HCV-induced HCC is the fact that the RNA hepatitis C virus cannot integrate into the host genome in a way similar to HBV. On the contrary, most of the carcinogenic effects of HCV are mediated through the action of its viral proteins, which deregulate host cellular cycle checkpoints, resulting in DNA mutations in liver cells. Some effects of HCV proteins are similar to those produced by the HBx protein of HBV [100,134,138,139].

Earlier reports showed that the HCV core and NS5A proteins play a critical role in HCC development [140,141,142]. Thus, the association of NS5A and p53 allows the transcriptional repression of the p21/waf1, a downstream effector gene of p53, and may contribute to HCV-mediated HCC [143]. HCV core and NS3 proteins can also activate TERT, the enzyme responsible for the length of the telomere. Short telomeres lead hepatocytes to apoptosis. Therefore, active TERT reduces apoptosis, favoring HCC. Indeed, increased TERT activity was associated with the aggressiveness and a poor prognosis of HCC [144]. HCV infection activates the Wnt/β-catenin pathway, leading to the subsequent activation of cell survival genes. The core protein reduces the expression of the Wnt antagonists [145,146]. The NS5A protein activates PI3K/Akt signaling pathway, reducing the degradation of β-catenin [147,148] and blocking apoptosis [149,150]. The c-Myc oncogene is also activated through the Wnt/β-catenin pathway in a murine mode [151]. The activation of the PI3K/Akt/mTOR pathway by NS4A is similar to the activation of the same pathway by the HBx protein of the HBV, both inducing HCC [152]. Moreover, the activation of the mTOR pathway was related to tumor differentiation and vascular invasion in HCC patients [153]. Lysosomes degrade the tumor suppressor p53 protein through the CMA autophagy (CMA). CMA is activated as a result of chronic ER stress and increased unfolded protein response related to HCV infection [154,155].

The viral proteins core and NS5A also affect lipogenesis [156]. HCV-infected cells had increased levels of polyunsaturated fatty acids (PUFAs) [157]. The accumulation of long-chain fatty acids in the infected hepatocytes finally leads to the activation of the NF-kB pathway, which leads to increased cellular survival and the development of HCC [158].

Finally, HCV proteins are implicated in an interplay of four signaling pathways, all of which are implicated in the induction and progression of HCC. EGFR is phosphorylated after the virus binds to its entry receptor, CLDN1/CD81. EGF pathway activation is sustained by the action of NS3/4A and maintained by the reduction in EGF degradation mediated by NS5A.The STAT3 pathway is activated through the direct action of the core protein and indirectly by the NS5A protein. The activation of the TGF-β pathway is mediated by intermediary of the UPR, and via the core protein. The VEGF pathway is activated by the active HIF-1a, which is activated by the core protein. The HCV core protein can trigger angiogenesis through a crosstalk between TGF-β2 and VEGF expression, thus favoring the progress of HCC [159]. Details on the four pathways interplay can be found in a recent detailed review [134].

The NS5B protein is also implicated in the induction and progression of HCC. HCV infection negatively regulates the retinoblastoma tumor suppressor protein (Rb). This is mediated by the NS5B protein, which is complexed with Rb, targeting it for degradation via the proteasome. The disruption of the Rb pathway in cells infected with HCV inhibits apoptosis and promotes chromosomal instability, factors that favor the development of HCC [160]. Some of these mechanisms may persist even after HCV eradication, and thus the risk of HCC development is not abolished. This is still a hotly discussed subject [154,161].

The mechanisms of HCC induction and progression in HCV infection have been recently reviewed in detail [3,162].

### 3.3. NAFLD

NAFLD is the third commonest risk factor of HCC in the United States. Initial events include genetic, metabolic, immunologic and endocrine pathways, which in turn activate oncogenic mechanisms [4,163,164,165].

TERT, CTNNB1, TP53 and ACVR2A are frequently mutated genes in NASH-associated HCC. Interestingly, ACVR2A tumor suppressor gene mutations were more common in NASH–HCC than in the HCC of other etiologies [166]. The patatin-like phospholipase-3 (PNPLA3) I148M sequence variant is the best genetic association of NAFLD/NASH to date and a strong risk factor for HCC [167]. The cycle-related kinase (CCRK) androgen-driven oncogene interacts with pro-inflammatory signals induced by obesity to promote NASH-related HCC [168]. However, the molecular profiles of NAFLD-associated HCC are heterogeneous and no discrete mutation profile was identified as it is the case in hemochromatosis and HCV-related HCC. By contrast, in HBV- and ALD-related HCC, distinct mutational signatures are usually identified [27,169]. However, a novel mutational signature was recently associated with NASH–HCC, but this finding requires further validation [166].

Epigenetic alterations similar to viral HCCs, such as DNA methylation, histone modifications and the silencing of microRNAs, were identified in NAFLD–HCC as well [170,171,172]. Some DNA methylation changes during NASH–HCC are different from those of viral-hepatitis-associated HCC. MAML3 is one among the DNA hypomethylated genes. MAML3 is a co-activator of β-catenin-mediated transcription, increasing the transcriptional activity of β-catenin [173]. Yet, the only epigenetic alteration that has clearly been linked to NASH-related HCC is the gene encoding chromodomain helicase DNA-binding protein 1 (CHD1) [174].

Specific microRNAs may also participate in NAFLD progression into HCC. MiR-301a upregulation and miR-375 downregulation were reported as HCC progresses from the early to late stages [175].

Inflammation and metabolic disturbances, such as diabetes, obesity and iron deposition, provide a favorable tumor microenvironment for the progression of malignant lesions. Over 90% of HCC occurs in association with liver inflammation [30]. Inflammation in NASH is similar to HBV and HCV, leading to the upregulated release of pro-inflammatory cytokines, such as IL-18, IL-1β, TNF-α and IL-6. TNF-α is probably the best studied pro-tumor cytokine in HCC. It activates the NF-kB and JNK signaling pathways to promote cell survival and inhibit apoptosis [176,177,178]. IL-6-mediated STAT3 activation is also a major driver of hepatocyte repair and replication, favoring HCC development [176]. Inflammation and hyperinsulinism in NASH are constant proliferative signaling mechanisms, which cause rapid HCC growth [179]. However, the sequence fatty liver-inflammation–NASH–fibrosis–HCC is not always linear, and some patients may progress from fatty liver to advanced fibrosis and HCC in the absence of significant inflammation [180,181,182].

Insulin resistance and hyperinsulinemia activate the PI3K-Akt and MAPK pathways to induce cell proliferation and inhibit apoptosis [165,183]. Immune and endocrine mediators originating from the gut microbiome provide an additional molecular mechanism that is implicated in HCC progression [184].

Oxidative stress is another important factor in HCC development in NASH. Hepatocytes overloaded with fatty acids generate ROS and ER stress as a result of mitochondrial dysfunction to cause cell damage predisposing to HCC [185,186,187]. An early event in NAFLD, associated with oxidative stress, is the promotion of pathological polyploidization and may participate in HCC development [188]. Iron deposition in the liver is an important inducer of oxidative stress in NASH. Elevated levels of iron are observed in NASH patients and are associated with HCC development [189]. Interestingly, oxidative stress, associated with iron overload in NASH, activates Wnt/β-catenin signaling and triggers carcinogenesis [190].

It should be noted that HCV and NASH have common metabolic abnormalities, such as hepatic steatosis, insulin resistance and oxidative stress. However, the underlying mechanism is different. The metabolic deregulation of HCV is induced by the core protein, in contrast to the complicated metabolic abnormalities of NASH [191]. The important contribution of apoptosis and autophagy in NASH-related HCC is examined later.

### 3.4. Diabetes

Type 2 diabetes mellitus (T2DM) predisposes to HCC even after adjustment for the presence of alcoholism, obesity and chronic viral hepatitis [192,193,194]. Moreover, the HCC recurrence rate is 2.5–4-fold higher in patients with T2DM, independently of the presence of cirrhosis or of the etiology of the liver disease [195,196]. Only in patients with chronic hepatitis B or primary biliary cholangitis, diabetes did not increase the risk of HCC [197]. However, in a nationwide Japanese study, the annual incidence of HCC in diabetes increased from 0.11% to 1.0% when advanced fibrosis was present [198]. Mechanistically, insulin resistance (IR) results in hyperinsulinemia as well as the activation of the insulin receptor and insulin-like growth factor 1 (IGF-1) signaling pathways, which are important initiators and supporters of hepatocarcinogenesis. HCC cells overexpress IGF-1 and insulin receptor substrate-1 (IRS-1). IGF-1 inhibits apoptosis and favors, therefore, HCC cell proliferation [4]. IRS-1 increased activity results in the activation of several cytokine pathways, including PI3K/AKT/mTOR, which modify cell cycle and favor cellular proliferation. IRS-1 also seems to prevent TGF-β-mediated apoptosis. In addition, alterations in lipid and glucose metabolism stimulate the production of ROS and cause mutations in the p53 onco-suppressor gene [199,200]. Moreover, data show that LINC01572 is upregulated in HCC tissues from patients with diabetes. The overexpression of LINC01572 increased HCC cell proliferation through sponging miR-195-5p, leading to an increase in glycolysis and the activation of the PI3K-AKT signaling pathway [201].

### 3.5. ALD

In most European countries, ethanol participates in the development of HCC be-tween 30 and 50% [202]. In France, the geographical distribution of HCC is not uniform, but the most affected regions are areas with a high wine production or those with excessive alcohol consumption [203]. On the contrary, in Crete, Greece, hepatocellular carcinoma is associated with the dispersion of HCV and HBV. ALD-related HCC was not very common in the past, but an increasing trend has been identified [204].

Hepatocarcinogenesis in excessive alcohol consumption is mostly due to the metabolic mechanisms associated with ethanol metabolism into acetaldehyde by alcohol dehydrogenase (ADH) and the microsomal CYP2E1. Acetaldehyde enters the mitochondria and oxidizes to acetate by mitochondrial aldehyde dehydrogenase (ALDH) [205]. There are two main mechanisms of cellular damage caused by acetaldehyde. The first is the formation of DNA and protein adducts. The second is the production of increased amounts of ROS by mitochondria, causing oxidative stress through lipid peroxidation that further deteriorates DNA mutagenesis. Oxidative stress is further aggravated by the iron deposition and lipid accumulation associated with excess ethanol [206]. ROS accumulation damages cellular macromolecules and is a critical factor in the progression of hepatocarcinogenesis through the formation of lipid peroxides, such as 4-hydroxy-nonenal [207].

An additional important factor, participating in liver carcinogenesis, is the increased gut permeability and the bacterial overgrowth caused by ethanol metabolism in the gut [208]. Endotoxins from the gut enter the portal vein and activate Kupffer cells, interacting with the receptor TLR4, which leads to the secretion of pro-inflammatory cytokines, such as IL-1, IL-6 and TNF-a, and to the initiation of liver inflammation [209,210]. NF-kB, one of the regulators of the inflammatory response, is also activated by TNF-a [211,212]. Moreover, the IL-6/STAT3 and TNF-a/NF-kB pathways have been implicated in hepatocarcinogenesis [213]. The mitogenic activity of hepatocyte is increased and hepatocyte apoptosis is inhibited, resulting in the induction of HCC [211].

The role of inflammasomal activation has been investigated in mice with ALD. IL-1β signaling is mandatory for the development of alcohol-induced liver steatosis, inflammation and injury. The upregulation of caspase-1 activity and inflammasome activation are the mediators of increased IL-1β [214].

Acetaldehyde production in the hepatocyte is influenced by the genetic variants of ADH and ALDH. The alleles ADH1C*and ALDH2*2 are associated with an increased probability of HCC [215,216,217]. Other genetic determinants are implicated in the severity of ALD, such as the patatin-like phospholipase domain-containing protein 3 (PNPLA3), the transmembrane 6 superfamily member 2 (TM6SF2) and the membrane-bound O-acyltransferase domain-containing protein 7 (MBOAT7) [218,219,220].

Moreover, acetaldehyde interferes with methyl group transference, leading to DNA hypomethylation and modifications of both oncogenes and tumor suppressor genes [221,222].

### 3.6. Hemochromatosis

Iron overload is the characteristic of hereditary hemochromatosis (HH), and increased iron produces increased ROS through the Fenton reaction, leading to DNA damage and HCC. Studies on the association of hemochromatosis and ferroptosis as a risk factor of HCC are limited, possibly because these studies were conducted before ferroptosis was described [223].

A recent study demonstrated that ferric citrate triggers ferroptosis in cells, suggesting the involvement of ferroptosis in HH [27]. This study also showed that SLC7A11 is a candidate gene of ferroptosis in HH and indicated that the Nrf2 activation may be a compensatory mechanism to protect against iron-overload-induced ferroptosis in HH [224].

Nonetheless, the risk of HCC in HHH was clearly overestimated in the past. More recent studies have indicated that this risk is lower and mostly occurs in patients with cirrhosis at the time of diagnosis. The true incidence of HCC in HH is better derived from population-based studies [225].

In a population study, the overall standardized incidence ratio of HCC was 1, which increases among first-degree relatives of the patients [226].

However, a very recent study suggested that HH without cirrhosis is an independent risk factor for HCC after adjustment for all known risk factors. The aOR was 28.8 higher than any other disease risk factor for HCC [227].

## 4. Apoptosis

Apoptosis is one of the forms of programmed cell death, in which characteristic cellular contents are not liberated into the surrounding environment. Apoptosis is mediated by a sequential activation of a series of caspases. The initiator 8 and 9 caspases are activated from pro-caspases upon sensing the initial signal via intracellular sensors, and activate the executioner 3, 6 and 7 caspases. The intrinsic pathway initiates apoptosis by an internal cell damage, while an external signal initiates the extrinsic pathway of apoptosis [228]. External stimuli, such as TNF-α, Fas ligand and TNF-related apoptosis-inducing ligand (TRAIL), operate through surface death receptors, while intrinsic stimuli operate by the mitochondrial signaling pathway [229,230]. The intrinsic pathway is associated with mitochondrial outer membrane permeabilization (MOMP), followed by the release of cytochrome c and leading to apoptosome formation. The effector caspases cleave hundreds of cellular proteins causing DNA fragmentation and actin reorganization, leading to membrane blebbing. Phosphatidylserine (PS) molecules exposed on the plasma membrane act as “eat me” signals for macrophages [11].

MOMP is regulated by the BCL-2 family. The pro-apoptotic BCL-2 proteins, BCL-2-associated X protein (BAX) and BCL-2 antagonist killer 1 (BAK) are activated by the pro-apoptotic proteins BAD and BID, leading to the activation of the caspase cascade and cell apoptosis. Protection from apoptosis is provided by pro-survival BCL-2 proteins [231].

This intrinsic apoptotic pathway initially increases the activity of pro-apoptotic BH3-only proteins that bind and neutralize the members of the pro-survival BCL-2 family [232,233]. BAK and BAX are then free to assemble into structures that cause MOMP [234,235]. Details of the apoptotic pathway have been recently reviewed [236,237,238].

### Apoptosis and HCC

In human HCC, the activation of the anti-apoptotic BCL-xL is usually associated with a parallel downregulation of BAX [239]. Moreover, the inhibition of caspases is also common in HCC, associated with TGF-β signaling. All these contribute to liver cancer initiation and progression [240]. As mentioned above, a significant number of HCC patients have alterations of the NF-kB pathway, particularly patients with NASH-induced HCC [241,242]. NF-kB activation via TNF promotes HCC by inhibiting apoptosis. The effects of NF-kB may promote HCC development by either activation or inhibition. The opposing different effect of NF-kB can be explained. The activation of NF-kB is linked to several pathways inhibiting apoptosis, such as the Bcl-2 family members and FLIP, but also to other pro-survival pathways [231,243]. On the other hand, increased hepatocyte apoptosis is associated with increased compensatory proliferation, leading to an increased incidence of oncogenic mutations. Murine experiments with reduced hepatocyte NF-kB activation increase hepatocyte apoptosis and compensatory proliferation, followed by increased predisposition to HCC [30,244]. Similar to NF-kB, c-Jun N-terminal kinase (JNK) can promote HCC development by inducing inflammation and hepatocyte proliferation, but it may also have an anti-tumorigenic function. Taken together, these findings indicate that the TNF-α, NF-kB and JNK pathways may have either pro-survival or cell death effects, both leading to HCC development [178,245,246,247].

The activation of caspases and other apoptosis-related molecules is a common finding in the liver of NASH patients [244]. Apoptotic hepatocytes stimulate immune cells and hepatic stellate cells, contributing to the progression of fibrosis. Inflammasomes, oxidative stress and ER stress also contribute to the progression of NASH and development of HCC by the induction of apoptosis [248], but the exact interplay is still debated.

Apoptosis may also be involved in the development of ALD-related HCC. JNK modulation has a dual role. Experiments in hepatocytes indicate that JNK activation by ethanol or acetaldehyde can be both pro- and anti-apoptotic. The activation of p42/44 MAPK, on the other hand, is anti-apoptotic, for both ethanol and acetaldehyde [249].

## 5. Autophagy

The understanding of the mechanism of autophagy ( a Greek word, meaning self-eating) is based on the pioneer works of Christian De Duve and Yoshinori Oshumi [250,251].

The sequential stages of autophagy include induction, phagophore, autophagosome and autolysosome formation and finally degradation [252,253,254].

The first step in the induction of autophagy is the formation of the ULK1 complex from the assembly of the ULK1, ATG13, FIP200 and ATG101 proteins. The ULK1 complex induces the formation of the PI3KC3 complex containing the proteins Beclin1, Atg14, VPS15 and VPS34. Both complexes are necessary for the formation of the autophagosome. Beclin1 regulates the effects of the complex. When the anti-apoptotic protein Bcl-2 binds to Beclin1, it reduces the affinity of Beclin-1 for VPS34 and inhibits autophagy. Beclin1 is released from Bcl-2 by BNIP3, another member of the Bcl-2 family with a BH3 domain, and autophagy is initiated. The protein Rubicon also binds to Beclin1 and inhibits the PIK3C3 activity. The transformation of phagophores into autophagosomes requires the Atg12–Atg5–Atg16 complex and the phosphatidylethanolamine (PE)-conjugated LC3II (Atg8) system. Finally, the autophagosome fuses with the lysosome for the degradation of the contents, which thereby degrades the autophagosomal contents (Figure 1) [255,256].

AMPK, a sensor of the cellular energy, is an important regulator of autophagy. Upon energy starvation, the activated AMPK initiates autophagy by increasing ULK1 activity through the serine phosphorylation of ULK1. Autophagy is inhibited by the PI3K/AKT/mTORC1 pathway, when enough cellular energy is available. AMPK can negatively regulate mTORC1, either directly through the phosphorylation of mTORC1 activity or indirectly by activating TSC2, which is a strong inhibitor of mTORC1 [257]. Recently, an additional mechanism for mTORC1 activation under energy-rich conditions was described. mTORC1 phosphorylates the protein Pacer, causing the disruption of the Pacer, Stx17 and HOPS complex, thus abolishing the autophagosome maturation mediated by this complex [258].

p38 also upregulates autophagy by inhibiting mTOR, while JNK and BNIP3 disrupt the Bcl-2–Beclin11 interaction, thereby initiating autophagy [259,260,261].

Two additional autophagy regulators have been described. The lncRNA NBR2 inhibits Beclin 1-dependent autophagy and suppresses autophagy-induced cell proliferation in HCC [262], while Forkhead box O3 (FOXO3), a member of the FOXO subfamily of transcription factors, upregulates autophagy, acting on ULK1, Beclin-1 and LC3 [263].

A detailed overview of the autophagy mechanisms involved in HCC development and progression have been recently published [264,265].

Mitophagy is a special form of autophagy that clears damaged mitochondria and is mediated by two molecular pathways. The first pathway is activated by the HIF1A/HIF-1a hypoxia-inducible factor 1 subunit alpha (HIF1A/HIF-1a). The second pathway is the PINK1 (PTEN-induced kinase 1)-PRKN (parkin RBR E3 ubiquitin protein ligase) pathway, activated by membrane depolarization. An important regulator of mitophagy is the TP53/p53, which facilitates mitochondrial dysfunction and disturbs the clearance of damaged mitochondria by mitophagy [266].

### Autophagy and HCC

Autophagy is implicated in the initiation and progression of HCC in many ways. It is closely associated with inflammation, which is a critical factor in HCC development. Autophagy and inflammasomes are interconnected as the same mechanisms regulate them, but through different pathways. The NLRP3 inflammasome activated by DAMPS induces caspase-1, leading to pyroptosis, as mentioned above [34,35].

Caspase-1 is also a mediator of autophagy activation. Autophagy eliminates inflammasomes and also damaged cellular organelles that would otherwise act as DAMPS [267,268]. However, this negative correlation between autophagy and inflammasomes is not always operative, as both can move towards the same direction in cases of NF-kB activation [269]. Moreover, the behavior of autophagy depends on the involvement of liver cells. Autophagy is protective in NAFLD and ALD, reducing lipid accumulation and oxidative stress. Autophagy activation in Kupffer cells also inhibits inflammation and liver fibrosis, but favors fibrosis if activated in HSCs [270,271].

In the early stages of cancer, autophagy behaves as a tumor suppressor, eliminating damaged mitochondria and unfolded proteins. It also decreases lipid accumulation in liver cells, reducing inflammation. Autophagy, however, acts as a tumor promoter after HCC induction, maintaining oxygen homeostasis to help the survival of malignant cells. In addition, it favors the appearance of resistance to treatment [272,273,274,275]. Both macroautophagy and CMA act as a double-edged sword in liver hepatocarcinogenesis, as shown by experimental and clinical studies. Mice with defective autophagy do not develop HCC, irrespective of any challenge, due to the activation of tumor suppressors, such as p53. However, after the induction of HCC, autophagy is necessary to degrade tumor suppressors, thus promoting the progression of HCC [276,277].

Increased levels of the autophagy marker LC3-II are correlated with lymph node metastasis, high vascular invasion and, most importantly, the reduced 5-year survival of HCC patients [278,279].

The macroautophagy flux is impaired in the final stages of HCC. However, during the later stages of HCC, more than 95% of tumors have an expression of LAMP-2A that is consistent with the induction of CMA in HCC [280]. CMA is probably upregulated under continuous, severe stressful stimuli and functions as a potential compensatory mechanism to reduce macroautophagy after the induction and establishment of HCC [281].

The activation of the Wnt/β-catenin pathway favors the development of HCC, as previously mentioned. Experimental evidence indicated that Wnt/β-catenin inhibitors repress the proliferation of HCC cells by regulating autophagy [282]. However, another report suggested that other mechanisms not related to autophagy led to an interference with Wnt secretion and a reduction in tumor growth through alternative, not-yet-identified pathways [283].

A special form of autophagy that removes damaged mitochondria is also a double-edged sword in HCC growth. Increased mitophagy was reported to suppress HCC cell survival [284,285]. The opposite has also been suggested, as increased mitophagy may promote hepatoma cell survival either through an increased production of ROS or through the attenuation of p53 activity [286,287].

Certain points should be noted in connection to the HCC of specific etiology. In HBV, the HBx protein increased autophagosome formation and reduced lysosomal acidification and the accumulation of immature cathepsin D [288,289]. This repression of lysosomal acidification is important for the development of HBV-associated HCC [290]. The inhibition of lysosomal degradative function by hydroxychloroquine induced p53 and increased apoptosis, but the activation of autophagy using the Torin-1 inhibitor of mTOR increased HCC growth [280]. Moreover, Arrestin beta 1 (ARBB1) promoted HCC formation through the interaction of HBx with LC3 and the promotion of autophagy [291]. HCV generates cellular stress and activates CMA autophagy to promote cell survival. CMA activation leads to HCC induction due to the repression of hepatic innate immunity and the degradation of several tumor suppressors [292].

Lipolysis and autophagy are interconnected. Autophagy reduces lipid accumulation, oxidative stress and inflammation in the liver, but autophagy also regulates adipogenesis and differentiation in the adipose tissue [293]. Similar to HBV, autophagic flux and the level of mature cathepsin D were reduced in three murine models of NAFLD, suggesting defective lysosome acidification under endoplasmic reticulum stress [294]. In NASH and NASH–HCC, autophagy has a dual role. On the one hand, autophagy reduces intracellular lipid droplets, attenuating lipotoxicity and inflammation. On the other hand, autophagy also affects adipogenesis and adipocyte differentiation. Basal autophagy, therefore, behaves as a tumor suppressor. After the induction of HCC, unbalanced autophagy contributes to carcinoma cell survival [295,296]. However, the contention of whether autophagy favors or inhibits NASH progression has not been settled. Defective autophagy is also linked to NASH–HCC through the induction of pro-inflammatory NF-kB activity, while defective mitochondria are retained, producing ROS to damage cellular DNA [297]. Interestingly, under conditions of reduced autophagy, hepatocytes were found to release the high-mobility group box 1 (HMGB1) protein, driving the proliferation of isolated hepatic progenitor cells. This could be an additional mechanism for the development of NAFLD–HCC [298].

ALD-associated HCC is also related to autophagy. Mitochondrial aldehyde dehydrogenase (ALDH2) is a critical enzyme further metabolizing the acetaldehyde produced by ethanol metabolism. Experiments in ALDH2 transgenic mice demonstrated that ALDH2 mitigates alcohol-induced liver steatosis and inflammation through the regulation of autophagy [299]. Moreover, TNF-α-induced protein 8 (TNFAIP8) is involved in the progression of HCC. TNFAIP8 induces autophagy by inhibiting the AKT/mTOR pathway in HCC cells. In addition, a direct interaction with ATtg3–Atg7 proteins was also reported. This mechanism is operative in the ALD of mice and humans, but not in NASH [300].

An important aspect of HCC development is the effect that autophagy exerts in the tumor microenvironment and particularly in TAMs. The increased autophagy of TAMs leads to the anti-tumoral M1 polarization, while the inhibition of autophagy leads to M2 polarization that favors hepatocarcinogenesis. The activation of the mTOR pathway, which is a negative regulator of autophagy, leads to M2 phenotype polarization and the promotion of HCC. The coagulants tissue factor (TF) and factor VII (FVII), locally produced in tumor microenvironment, promote HCC growth by the repression of autophagy mediated by mTOR activation and Atg7 [301,302].

## 6. Interplay between Apoptosis and Autophagy

Autophagy and apoptosis are normally tumor suppressor pathways. The degradation of oncogenic molecules by autophagy prevents cancer initiation, while apoptosis eliminates cancer cells. Under conditions of stress, autophagy may facilitate the survival of tumor cells [303]. Similar external or internal signals can induce either apoptosis or autophagy. They usually exhibit mutual inhibition, although single-cell experiments indicated that, in many instances, they are both operational. Apoptosis and autophagy may act in concert to kill or, alternatively, the activity of one mechanism can exclude that of the other. The result is important for the effectiveness of chemotherapy in several cancers, including HCC [304]. Usually, autophagy precedes apoptosis [305]. The initial activation of autophagy is an effort towards survival. The initiation of apoptosis will eventually kill the cell if autophagy fails. The induction of autophagy inhibits apoptosis, while apoptosis suppresses autophagy initiation [306]. Bcl-2 is an important regulator of the interplay. Bcl-2 inhibits the pro-apoptotic Bax and interacts with the PI3K complex of the autophagy pathway, promoting survival. However, the phosphorylation of Bcl-2 inhibits its binding to Bax, leading to apoptosis [307]. A pro-apoptotic role of autophagy has also been reported [308].

An example of the concerted action of the two pathways to inhibit the replication of hepatocellular carcinoma cells was recently published. Solamargine, a traditional Chinese herb medicine, induced both apoptosis and autophagy to repress the replication of hepatoma cell lines [309]. Similarly, Jujuboside B, an ingredient of the traditional Chinese medicine Zizyphi Spinosi Semen, induced both autophagy and apoptosis in breast cancer cells [310]. A synergy between autophagy and apoptosis was also described in the anti-fibrotic activity of curcumol. It can induce both the autophagy and apoptosis of hepatic stellate cells. Since fibrosis is an important factor for the initiation of HCC, this dual action of curcumol may favorably influence HCC initiation [311]. However, a detrimental outcome may be the result of the concerted action of apoptosis and autophagy. This was reported in human kidney mesangial cells incubated with homocysteine, which induced ER stress. Both autophagy and apoptosis were activated, and the viability of cells was significantly reduced [312].

On the other hand, the ER stress and UPR that follows is an example of the mutual inhibition of apoptosis and autophagy. The accumulation of unfolded proteins in the ER lumen induces ER stress and the activation of three major UPR pathways (PERK, IRE1α and ATF6) leading to UPR. The final result is the inhibition of apoptosis and the activation of autophagy. This mechanism may be related to the proliferation of HCC cells and the resistance of HCC to chemotherapy [47]. The impact of autophagy on cell survival during ER stress varies according to the tissue type. ER-induced autophagy protects against cell death in colon and prostate cancer cells. However, in normal human colonocytes, autophagy does not counteract ER stress but facilitates ER-induced apoptosis [313]. A mutual exclusion is not operative only in the liver. It is operative in secondary hyperparathyroidism cells, where the autophagy inhibitor chloroquine enhances experimentally induced apoptosis [314]. In hepatocellular carcinoma, autophagy may either support apoptosis or antagonize apoptosis. The activation of autophagy may lead to the induction of apoptosis and the inhibition of the growth of hepatoma cells [315,316]. Experimental evidence indicates that lipophagy, a special form of autophagy, can also act in both ways. It can either supply tumor cells with energy important for their proliferation or suppress tumor development through the direct inhibitory effect of acid lipase [317,318]. Lipophagy can also induce apoptosis via the induction of mitochondrial stress [319].

Experimental evidence has also indicated that an interplay between autophagy and apoptosis may be implicated in the pathogenesis of NASH and ALD. JNK1 increases palmitate-induced lipoapoptosis, whereas JNK2 activates autophagy and inhibits palmitic acid lipotoxicity, improving the survival of hepatoma cells [320,321]. The promotion of autophagy by the mitochondrial uncoupling protein 2 (UCP2) also inhibits apoptosis [322]. The inhibition of autophagy by the tumor protein p53-binding protein 2 (TP53BP2) may be involved in NASH [323]. The overexpression of Rubicon, a Beclin-1-interacting negative regulator for autophagosome–lysosome fusion, causes the suppression of the late stage of autophagy. Its blockade mitigated autophagy suppression and reduced palmitate-induced ER stress and apoptosis [324]. Parkin-mediated mitophagy may attenuate apoptosis, improve the quality of mitochondria and suppress hepatocyte steatosis in models of ALD due to Parkin translocation into mitochondria [325]. Sirtuin 3 (SIRT3) is a nicotinamide adenine dinucleotide-dependent deacetylase located within the mitochondria. SIRT3 is a negative regulator of autophagy. SIRT3 overexpression causes AMPK inhibition, mTOR activation and finally autophagy suppression, promoting the hepatocyte lipotoxicity induced by saturated fatty acids [326].

An important field of research is the identification of pathways where autophagy and apoptosis meet (Figure 2). Several pathways that mediate the interplay between autophagy and apoptosis have been identified and are analyzed in this paper [327].

### 6.1. Beclin-1

The Beclin-1/BCL-2 interaction was the first described molecular connection between autophagy and apoptosis. Beclin-1, the mammalian homolog of the yeast Atg6, participates in autophagosome formation as a component of the PI3K complex [328,329]. The interplay between autophagy and apoptosis is mediated, in part, by the interaction between Beclin-1 and the anti-apoptotic proteins BCL-2 and BCL-XL [330,331], as previously mentioned. This inhibits the pro-autophagic function of Beclin-1, but does not interfere with the anti-apoptotic activity of the BCL-2 family proteins. In addition, the inactivation of Beclin-1 triggers apoptosis [331]. Several BH3-only proteins can activate both autophagy and apoptosis. To induce apoptosis, BH3-only proteins directly neutralize anti-apoptotic proteins from the BCL-2 family, and stimulate those with pro-apoptotic functions. Beclin-1 is such a protein, as it possesses a BH3 region [332]. inhibiting these anti-apoptotic proteins, or, alternatively, activating the pro-apoptotic BCL-2 family members, such as BAX and BAK [333]. On the other hand, BH3-only proteins disrupt this interaction and permit Beclin-1 to increase autophagic activity. Only BIM, a unique BH3-only protein, has an opposite effect on autophagy. BIM interacts with Beclin-1 and prevents autophagy. NIX, another BH3-only protein, localized in the mitochondria, favors mitophagy. JUN N-terminal kinase (JNK) is also associated with autophagy regulation. JNK induces autophagy or apoptosis through the phosphorylation and inactivation of BCL-2, leading to apoptosis or through the phosphorylation of BIM that disrupts the inhibitory interaction with Beclin-1, leading to autophagy [306].

### 6.2. Beclin-1 in HCC

Autophagy is significantly reduced in the most aggressive HCC cell lines and tissues, particularly when the Bcl-xL protein is overexpressed. These findings were corroborated in curative resection specimens from HCC patients where the reduced expression of Beclin-1 was negatively correlated with survival only in the Bcl-xL+ patients, indicating that an increased expression of the anti-apoptotic gene Bcl-xL was associated with decreased expression of Beclin-1 and a poor prognosis [334].

These results were verified in two additional studies. The first was performed in material from 103 HCC patients, where Beclin-1 was negatively correlated with the anti-apoptosis protein Bcl-2 and positively correlated with the pro-apoptosis protein Bax. The 5-year survival rates were considerably higher among patients with strong Beclin-1 positivity compared to those with weak or negative expression [335]. The second study of 35 HCC patients reported similar results [336].

Interestingly, it was found that nitric oxide (NO) may influence the autophagy–apoptosis balance in HCC through Beclin-1. The levels of NO were significantly increased in HBV-related HCC compared to cirrhosis. Further experiments with human hepatoma cells showed that NO induced apoptosis and inhibited autophagy, whereas the induction of autophagy could attenuate NO-induced apoptosis. NO controls the switch between apoptosis and autophagy, disrupting the Beclin-1/Vps34 complex and increasing the Bcl-2/Beclin-1 connection [337].

The actions of sorafenib, a drug used for the treatment of advanced HCC, are additional evidence for the significance of Beclin-1 in regulating autophagy and apoptosis. Sorafenib induces autophagic cell death in HCC through Beclin-1 and apoptosis [338]. The induction of apoptosis by sorafenib is probably a more important mechanism for hepatoma cell death as the inhibition of autophagy augments the effect of sorafenib, increasing apoptosis [339,340].

A very recent report used a different approach that showed the interplay between autophagy and apoptosis. Vaccinia-related kinase 2 (VRK2) increases sorafenib resistance in HCC cells. This is obtained by the phosphorylation of Bcl-2, thus enhancing the dissociation of Bcl-2 from Beclin-1, followed by the formation of the Beclin-1/Vps34 complex, which facilitates autophagy. Furthermore, VRK2 phosphorylated Bcl-2, promoting the interaction of Bcl-2 with BAX, thereby reducing apoptosis [341].

In addition, Beclin-1 is involved in the regulation of apoptosis through the action of caspase. Growth factor depletion leads to the caspase-mediated cleavage of Beclin-1, impairing autophagy. A fragment of Beclin-1 is then generated and localized to mitochondria, leading cells to apoptosis through the release of pro-apoptotic factors, such as BAX [342]. The pro-apoptotic protein BAX reduces autophagy, promoting the caspase-mediated cleavage of Beclin-1. This is an indication that apoptosis can suppress autophagy [343]. The link between autophagy and apoptosis is further supported by evidence that other autophagy-related proteins, such as ATG5, are also substrates for caspase cleavage and the induction of apoptosis. The cleaved ATG5 translocates into the mitochondria, inducing the mitochondrial apoptotic pathway [344]. Therefore, the caspase-mediated cleavage of ATG5 and Beclin-1 switches autophagy to apoptosis. The involvement of caspase-3 constitutes a switch between autophagic or apoptotic cell death [345].

### 6.3. mTOR Interaction with Autophagy–Apoptosis and the Regulation of mTOR in HCC

mTOR is implicated in several signaling pathways regulating cell proliferation, autophagy and apoptosis [346]. There are two main mTOR signaling pathways: the classical PI3K/Akt/mTOR and the LKB1/AMPK/mTOR signaling pathways. The glycogen synthase kinase 3 beta (GSK3B)-mediated phosphorylation of ULK1 is important in autophagy induction, suppressing the mTOR pathway and potentially inducing tumorigenesis [347,348].

mTOR has also several effects on apoptosis depending on the cells involved and its effect on the activation of downstream targets, such as p53 and BCL-2 proteins [349].

The anti-apoptotic BCL-2 homolog MCL1 controls autophagy and apoptosis. The interplay between BAX and Beclin-1 downstream of MCL1 degradation finally determines if autophagy or apoptosis will prevail. It should be noted that mTOR inhibition, following nutrient deprivation, causes MCL1 degradation [350]. On the contrary, both autophagy and apoptosis may be controlled through the activation of the mTOR pathway. Thus, β-carotene attenuated both the apoptosis and autophagy of enterocolitis IEC-6 cells stimulated with LPS, activating the PI3K/AKT/mTOR signaling pathway [351]. There is strong experimental evidence that the mTOR pathway regulates autophagy and apoptosis in HCC. The importance of the PI3K/Akt/mTOR signaling pathway in HCC induction and progression has been established. It is implicated in every etiology of HCC (viral, ALD and NASH). The mTOR pathway is overexpressed in almost 50% of HCC and the impaired activation of this pathway affects cell proliferation, differentiation, autophagy and the epithelial–mesenchymal transition (EMT) [352,353].

Apigenin, a dietary flavonoid, induced apoptosis and autophagy in HCC cells by inhibiting the PI3K/Akt/mTOR axis. Although autophagy protected cells from death, the end result was the inhibition of cellular proliferation [354]. Brusatol, a traditional Chinese herbal medicine, inhibited proliferation and induced apoptosis in liver cancer lines. The autophagy inhibitor chloroquine attenuated Brusatol-induced apoptosis, indicating that Brusatol promoted autophagy-induced apoptosis in HCC through the inhibition of the PI3K/Akt/mTOR axis [355].

The upregulation or downregulation of mTOR-related oncogenic lncRNAs contributes to the aberrant expression of oncoproteins, leading to the disturbed regulation of the mTOR axis [356,357]. The aberrant expression of lncRNAs is associated with the metastasis, recurrence and chemoresistance of HCC [358]. In particular, the inhibition of the lncRNA HIF1A-AS1 increases apoptosis by reducing HIF-1α/mTOR-induced autophagy, while its overexpression is related to the TNM stage and lymph node metastasis [359]. A synergistic effect of PI3K/AKT/mTOR pathway-induced autophagy and apoptosis was recently reported. The concomitant incubation of hepatoma cell lines with aloin and metformin inhibited cellular proliferation, increasing both autophagy and apoptosis [360]. The regulation of mTOR in HCC has been recently reviewed [361].

### 6.4. p27kip1

p27 kip1 is a cyclin-dependent kinase inhibitor and a tumor suppressor. p27Kip1 is a critical mediator of autophagy and apoptosis. Unlike other tumor suppressors, such as p53, the loss of p27 expression, frequently found in tumors, occurs via proteasomal degradation or re-localization, and not through genetic or epigenetic modifications [362]. The cellular location of p27Kip1 is partially controlled by phosphorylation from several kinases, such as Akt and AMPK. Thus, the cytoplasmic location of p27Kip1 has been found to promote cellular survival through autophagy (Figure 3).

Nuclear p27Kip1, however, increases cell susceptibility to apoptosis or senescence [363]. A reduction in energy metabolism activates the LKB1-AMPK energy-sensing pathway, leading to the phosphorylation and stabilization of p27kip1. Autophagy is induced and cell survival is increased. A reduction in p27kip1 under these conditions activates apoptosis [364,365]. Recently, the DNAJC5 protein was reported to be associated with the regulation of p27. DNAJC5 expression is frequently increased in human HCC and is strongly related to poor prognosis. DNAJC5 enhances the degradation of p27, while DNAJC5 knockdown reverses the decrease in p27 levels, indicating that the oncogenic function of this protein is p27-mediated [366]. A recent meta-analysis indicated that there was a significant correlation between low p27kip1 expression and aggressive progression, leading to a shorter overall survival in HCC patients [367].

### 6.5. The Anti-Apoptotic FLIP

The cellular FLICE inhibitory protein (c-FLIP) and the viral FLIP (vFLIP) are important anti-apoptotic proteins against death-receptor-mediated apoptosis and necroptosis [368]. There are three isoforms of c-FLIP: c-FLIPL (long form), c-FLIPS (short form) and c-FLIPR (Raji form). They all share the DED1 and DED2 domains [369].

Apoptosis is inhibited by FLIP through the interruption of the cell death machinery [370]. FLIP binds to procaspase 8, one of the molecules that is involved in apoptosis induction, and stops its maturation, inactivating thus the downstream apoptosis cascade [371]. However, the end result of FLIP implication depends on the level or type of c-FLIP isoforms involved. The c-FLIPL negatively regulates necroptosis, but the c-FLIPS promotes RIP3-mediated necroptosis [372]. The c-FLIP isoforms determine whether cell death follow either through the caspase-dependent apoptosis or through the RIP3-mediated necroptosis. Additionally, c-FLIP redresses autophagy, inhibiting Atg3-binding LC3, which is an essential component for autophagosome formation [373]. Therefore, FLIPs act not only as anti-apoptotic factors, but also as suppressors of autophagy. Moreover, a DED1 peptide or a DED2 peptide of FLIP effectively suppress the Atg3–FLIP interaction without affecting the Atg3–LC3 interaction, resulting in cell death. These FLIP-derived short peptides, therefore, induce growth suppression and cell death by autophagy [373].

FLIP is implicated in the many actions of the HBx protein. The pro-apoptotic function of HBx is mediated through its interaction with c-FLIP variants [374], thus being anti-viral. On the other hand, c-FLIP may be also pro-viral because it stabilizes HBx [375].

Associations between HCV viral proteins and c-FLIP were also described. The HCV core protein maintains the expression of c-FLIP, ultimately blocking TNFα-mediated apoptosis [376], but the opposite results were also reported as HCV core, NS4B and NS5B proteins enhance TNF-induced apoptosis. HCV proteins also reduced the expression of NF-kB-dependent anti-apoptotic proteins, such as Bcl-xL, and c-FLIPL [377]. The hedgehog proteins are implicated in the action of FLIP in HCC. The abnormal activation of the hedgehog pathway is associated with the occurrence of HCC. The protein Gli2 is a terminal transcription factor in this pathway. Gli2 downregulation enhanced TRAIL-induced apoptosis through the reduction in c-FLIP and Bcl-2, indicating the importance of Gli2 in the activation of c-FLIP. On the other hand, the increased expression of c-FLIP alleviated TRAIL-induced apoptosis via the suppression of caspase-8 [378].

### 6.6. The Role of the ATG12, ATG5 and ATG3 Proteins in Autophagy

ATG12 is an important mediator of the direction of the balance between autophagy and apoptosis. ATG12, in association with ATG3, inhibits the anti-apoptotic Bcl-2 and promotes apoptosis. When ATG12 is associated with ATG5, autophagy is increased. The calpain cleavage of ATG5 switches autophagy to apoptosis [305,344,379,380].

### 6.7. The Death-Associated Protein Kinase (DAPK) Family in Apoptosis and Autophagy

Death-associated protein kinases (DAPK) are members of a family of five related kinases that mediate several cellular pathways, including apoptosis, autophagy and tumor suppression. The three better-studied family members are DAPK1/DAPK, DAPK2 and DAPK3/ZIPK, which share a high degree of homology but different cellular localization [381]. Initial studies demonstrated that DAPK can induce apoptosis by several pathways, such as p53- and mitochondrion-dependent apoptosis in hepatoma cells [382,383]. However, the effect of DAPK2 in apoptosis is debatable. It seems that the overexpression of DAPK2 causes significant apoptosis but only in cancer cells detached from the extracellular matrix [381,384,385]. DAPK2 was shown to promote the initiation step of autophagy by decreasing mTORC1 activity [386]. DAPK2 is subsequently involved in the additional steps of autophagy. Beclin-1 is a target of DAPK. The DAPK-mediated phosphorylation of Beclin-1 promotes the dissociation of Beclin-1 from its inhibitor BCL-2 to induce autophagy [387,388]. SB203580 is an inhibitor of the p38 mitogen-activated protein kinase (MAPK) but also reduces cell proliferation in a p38/MAPK-independent way. This is achieved through the induction of autophagy in HCC cells associated with the activation of both AMPK and DAPK, which facilitates the phosphorylation of p53 and enhances Beclin-1 expression. The induction of autophagic death may, therefore, account for the antiproliferative effect of SB203580 in HCC cells [389]. Recently, the DEAD-box helicase 20 (DDX20) protein was identified as a downstream target of DAPK that leads to the tumor suppressor function of DAPK in HCC. DAPK1 ameliorated the proteasomal degradation of DDX20. DAPK also suppressed hepatoma cell migration and invasion, but not proliferation [390]. It should be noted that, in other cancers, an opposite effect may be observed. In human placental micro-vascular endothelial cells, DAPK2 overexpression led to a decrease in both autophagy and apoptosis connected to a decrease in Beclin-1 and BAX, along with an increase in Bcl-2 [391]. DAPK1 attenuated oxidative stress and reduced autophagy and inflammation by inhibiting the p38MAPK/NF-kB pathway in a mice model of acute lung injury [392]. In addition to autophagy and apoptosis, the precise role of DAPKs in HCC biology is not known. In a DAPK1 knockout model, hundreds of upregulated genes and downregulated genes were identified. The tissue metalloproteinase inhibitor 1 (TIMP1) and Alpha-2-HS-glycoprotein (AHSG) exhibited the strongest associations with DAPK1 elimination [393].

### 6.8. p53

The tumor suppressor p53 is encoded by the TP53 gene and is a critical regulator of autophagy and apoptosis in HCC. It is a sensor of cellular stress and responds to a variety of stimulants, such as DNA damage and oxidative stress [394]. It controls apoptosis by inducing the association of components of the extrinsic death receptor system [395] with several various mitochondrial pathways, such as PUMA and BAX, which in turn promote cell death [305,396,397,398,399]. Under stressful conditions, the cytoplasmic p53 translocates to the mitochondrial surface, promoting either the inhibition of the anti-apoptotic Bcl-2 family members or the activation of the pro-apoptotic members leading to the formation of pores in the mitochondrial outer membrane, cytochrome C release and apoptosis [400,401].

In contrast to apoptosis, the upregulation of cytoplasmic p53 or nuclear p53 has different effects in the regulation of autophagy. p53 exerts both pro- and anti-autophagic functions. This is dependent on its subcellular localization. The cytoplasmic p53 inhibits autophagy, acting on the UNC-51-like kinase 1 (ULK1) complex. Under stressful conditions, p53 translocates to the nucleus where it can promote autophagy by inhibiting mTOR through the activation of the AMP kinase [402,403] or the transactivation of the damage-regulated autophagy modulator (DRAM), which promotes the formation of autophagolysosomes [404]. The induction of autophagy via DRAM leads also to apoptotic cell death. Therefore, DRAM is an important element of the mechanism that controls p53-mediated apoptosis and autophagy [380,404]. In addition, nuclear p53 promotes the phosphorylation of Bcl-2. Phosphorylated Bcl-2 does not bind to Beclin-1, allowing the promotion of autophagy [380,405,406].

A different mechanism of the implication of p53 in autophagy and apoptosis regulation has been described. The high mobility group box 1 (HMGB1) and p53 form a complex that controls the balance between autophagy and apoptosis. The loss of p53 increased cytosolic HMGB1 expression and induced autophagy. On the other hand, the loss of HMGB1 increased cytosolic p53 and decreased autophagy. The effects on apoptosis were opposite. Therefore, p53 seems to be a negative regulator of the HMGB1/Beclin-1 complex, up- or downregulating autophagy and apoptosis [407].

The role of Krüppel-associated box (KRAB)-type zinc-finger protein ZNF498 in p53-induced apoptosis was recently reported in HCC. This protein suppressed apoptosis and ferroptosis by decreasing p53 phosphorylation in HCC development [408]. However, convincing evidence that p53 triggers apoptosis is available only for the wild-type. For instance, one study has shown that, in estrogen-positive breast cancer cells, the expression of a truncated p53 mutant increased BCL-2, thus decreasing their apoptosis in breast cancer cells [409]. Moreover, evidence has suggested that certain gain-of-function or loss-of-function mutations of the TP53 gene, as found in many cancers, turn p53 into an oncogene [410]. In this context, it should be considered that TP53 mutations are very common in hepatocellular carcinoma, and their interplay in the regulation of apoptosis and autophagy has not been investigated [411].

### 6.9. Tumor-Associated Macrophages (TAM) and the Tumor Microenvironment (TEM)

As previously mentioned, HCC, as most other cancers, have inflammation as a basic pathogenetic factor. TAMs play an important role in the maintenance of inflammation by producing several pro-inflammatory cytokines and chemokines [412]. The function of TAMs is regulated by autophagy [413]. Kupffer cells with autophagy deficiency promote liver inflammation and hepatocarcinogenesis via the production of ROS by the mitochondria [414]. TLR2 activation by hepatoma factors results in autophagy augmentation and the M2 immunosuppressive differentiation of TAMs [415]. TLR2-deficient mice had an unexpected increase in HCC induction and progression because TRL2 deficiency resulted in a decrease in macrophage infiltration and suppressed autophagy and apoptosis [416]. The natural compound baicalin shifted the differentiation of TAMs into the M1 anti-tumor phenotype and decreased hepatoma cell proliferation by increasing autophagy [417]. An interesting interplay of autophagy and a form of apoptosis called anoikis has been described. Cancer cell detachment from ECM induces cell death via anoikis. In the interplay between anoikis and autophagy, the ECM-integrin-activated dual tyrosine kinase complex of SRC is involved. SRC was demonstrated to regulate AMPK autophagy. When cells are detached, SRC is inactive, and AMPK is activated to induce protective autophagy against anoikis. When cells are attached again, SRC activation, reduces AMPK activity and downregulates autophagy, allowing cells to proliferate. Whether this mechanism is operative in HCC is not known at present [418].

TAMs are part of the tumor microenvironment that contains other immune cells, such as CD8+ T cells, T regulatory cells, myeloid-derived suppressor cells (MDSCs), dendritic cells (DCs), B cells and natural killer (NK) cells. These immune cells are regulated by similar signals and metabolic pathways with HCC cells. Therefore, this overlap makes them prone to similar vulnerabilities, with HCC cells making it difficult to attack only tumor cells without reducing antitumor immunity. Current research has produced conflicting results, but this is a very promising field that may exploit ferroptosis with immunotherapy in HCC treatment [419,420,421].

### 6.10. The Role of Mitochondria

Autophagy generates metabolic products, such as glutamine, to replenish TCA cycle intermediates that are used to sustain the mitochondrial metabolism of tumor cells, thereby sustaining mitochondrial metabolism in tumor cells [422]. In this context, chloroquine, a small-molecule inhibitor of autophagy, was shown to damage mitochondrial metabolism and diminish tumor growth [423,424,425].

Mitochondrial dysfunction promotes the accumulation of ROS, mtDNA damage and proto-oncogene activation, which are associated with the induction and progression of HCC [426,427]. A reduction in the mitochondrial membrane permeability (MMP) inhibits the apoptosis of HCC cells [428]. MicroRNAs targeting mitochondria showed that miR-518d-5p inhibits c-Jun/PUMA-induced apoptosis and increases sorafenib resistance in HCC [429]. On the contrary, the natural compound dehydrocrenatidine (DEC) reduced ATP production and disrupted the MMP of mitochondria in hepatoma cell lines. DEC induced mitochondrial impairment, increased apoptosis and exerted anti-tumor effects [430]. Interestingly, the inhibition of mitochondrial autophagy (mitophagy) induced the accumulation of damaged mitochondria in HepG2 cells and reduced both the proliferation of HCC cells and the resistance of HCC to sorafenib [431].

The role of mitochondria in tumor biology through the onset, maintenance and counteraction of apoptosis and autophagy has been recently reviewed [432,433].

### 6.11. Other Factors

There is evidence that Ca^2+^ regulates both autophagy and apoptosis, but the exact mechanisms are still unknown. An increase in Ca^2+^ induces autophagy but inhibits apoptosis, resulting in increased cell survival and proliferation. In theory, this is detrimental for HCC [14]. Activated ribosomes are associated with HCC. The RNA-binding protein PNO1 is an important ribosome in tumorigenesis. PNO1 was reported to be overexpressed in HCC, leading to autophagy promotion and apoptosis inhibition through the MAPK signaling pathway [434]. The interplay of autophagy and apoptosis in cancers has been recently reviewed [406].

### 6.12. Ferroptosis

Ferroptosis is an iron-dependent regulated cell death characterized by iron overload, lipid peroxidation and the overproduction of ROS [435,436]. The word derives from the Greek word “ptosis”, meaning a fall, and the Latin “ferrum”, for iron. Biochemically, ferroptosis is characterized by the consumption of glutathione (GSH) and the decreased activity of GPX4.

There are three main mechanisms regulating ferroptosis:(1)The glutathione/glutathione peroxidase 4 (GSH/GPX4) pathway, involving the system Xc−, which imports cystine and exports glutamate. A central role in this system is that of the cystine/glutamate exchanger solute carrier family 7 member 11 (SLC7A11) and the SLC3A2 exchanger [437,438,439].(2)Ferritinophagy and other iron metabolism pathways, particularly the p62-Kelch-like ECH-associated protein 1 (Keap1)-Nrf2 regulatory pathways.(3)The lipid metabolism pathways, implicating the tumor suppressor p53. p53 promotes the sensitivity to ferroptosis via the suppression of SLC7A11 [435,438].

Experimental evidence showed that ferroptosis is controlled by a variety of external inhibitors and activators [23]. Ferroptosis is initiated by a special form of autophagy called “ferritinophagy”, leading to the degradation of ferritin [440]. Several proteins involved in autophagy are also involved in ferroptosis. The elimination of Atg 5 and Atg7 reduced ferroptosis, induced by the ferroptosis activator erastin. The nuclear receptor coactivator 4 (NCOA4) is the selective carrier of ferritin to ferritinophagy. The genetic inhibition of NCOA4 reduces ferritin degradation and represses ferroptosis, while the overexpression of NCOA4 increases ferritin degradation and ferroptosis.

Ferroptosis is also induced by lipid peroxidation. The overexpression of ACSL4 is responsible for the synthesis of increased levels of polyunsaturated fatty acids (PUFAs), mainly from cell membranes rich in phospholipids, which promote ferroptosis. On the other hand, ACSL3 is responsible for the synthesis of monounsaturated fatty acids (MUFA) that induce ferroptosis resistance. Three mechanisms, namely, the cystatin–GSH–GPX4, the CoQ10–FSP1 and the GCH1–BH4–DHFR axes, all fueled by NADPH, can counteract ferroptosis by inhibiting lipid peroxidation [438].

Inducers of ferroptosis, such as erastin and sorafenib, act by two mechanisms. They inhibit the Xc--mediated cystine antiporter, reducing GSH and GPX4 and leading, therefore, to ROS accumulation, and ferroptosis induction. Another mechanism is related to the p62-Kelch-like ECH-associated protein 1 (Keap1)-nuclear factor E2-related factor 2 (NRF2) pathway. Nrf2 is a transcription factor that protects HCC cells from oxidative damage. p62 inhibits Keap1 and favors Nrf2 accumulation. Nrf2 activates retinoblastoma (Rb) and metallothionein (MT-1G) and induces ferritin heavy chain 1 (FTH1), quinone oxidoreductase 1 (NQO1) and HO-1. The administration of erastin or sorafenib leads to the upregulation of MT-1G and p62 and the downregulation of Rb (Figure 4) [223,435,441,442,443].

Beclin-1 was reported to increase ferroptosis by binding to SLC7A11. The elimination of Beclin-1 inhibits ferroptosis and is initiated by the system Xc- inhibitors, such as erastin, sulfasalazine and sorafenib. On the contrary, the activation of Beclin-1 promotes cancer cell death by ferroptosis, but not by apoptosis or necroptosis [444]. Autophagy can coincide with ferroptosis [445]. Ferroptosis was initially described as a separate type of regulated cell death, distinct from apoptosis and autophagy. However, it is now evident that autophagy, at least under certain conditions, contributes to ferroptotic cell death. Moreover, ferroptosis may share common signals or regulators with apoptosis [23,446,447].

### 6.13. Ferroptosis and HCC

Liver iron overload and ferroptosis have been conclusively linked to HCC initiation and progress [448,449,450]. As mentioned above, p53 is involved in the regulation of ferroptosis. A single-nucleotide polymorphism at codon 47 of TP53 leads to the disruption of p53 functions and resistance to ferroptosis, probably via the transcriptional regulation of SLC7A11 expression [451]. In general, genes may act as negative regulators of ferroptosis, increasing the resistance of HCC to drugs, such as sorafenib [452]. An increase in the expression of metallothionein-1G (MT-1G), which is a negative regulator of ferroptosis, increases resistance to sorafenib [453]. Ceruloplasmin also inhibits ferroptosis in HCC and increases the deposition of iron and ROS production [454]. In contrast, the synthetase long-chain family member 4 (ACSL4) is an essential mediator of ferroptosis execution and promotes ferroptosis in HCC (Figure 4) [455]. An upregulation of the ACSL4 protein in HCC tissues from responders to sorafenib has been demonstrated [456,457].

HBx causes the upregulation of ACSL4 by targeting miR-205, leading to the accumulation of cholesterol, and the development of HCC [458,459].

ACSL4 promotes the progression of HCC cells. The blocking of hexokinase H2 (HK2) activates ACSL4 effectively and leads to HCC progression [460,461].

Natural omega-3 PUFAs are important substrates in the induction of ferroptosis and the inhibition of tumor progression [462], a fact that can be exploited in HCC [463,464]. LncRNAs are also regulators of ferroptosis in HCC, but their action has not been clarified [465,466]. Recently, signature models using lncRNAs and ferroptosis were established, classifying HCC patients into two groups. The high-risk group had enhanced hepatocarcinogenesis and poor prognosis [467,468]. Equally, non-coding circular RNAs (circRNAs) are associated with the development of HCC through ferroptosis. Circ0097009 endogenous RNA controls the expression of SLC7A11 [469]. Novel ferroptosis-associated genes have been proposed for prognostic use in HCC [470,471]. Despite the all increasing importance of ferroptosis, there are no data on a possible interplay between ferroptosis, autophagy and apoptosis in HCC. The role of ferroptosis in HCC initiation and progression has been extensively reviewed [472,473].

## 7. Implications of Autophagy, Ferroptosis and Apoptosis in the Drug Treatment of HCC

Most patients with HCC are only candidates for drug treatments by the time they are diagnosed, as the tumor is unresectable or not suitable for loco-regional treatment [474].

Despite the introduction of several new drugs, the outcome is still unsatisfactory because resistance is rapidly developed. Interestingly a commonly used class of drugs may reduce the appearance of HCC. A meta-analysis demonstrated that statins may decrease HCC occurrence. This protection was more evident in HBV patients. Lipophilic statins, such as Atorvastatin, showed a greater effect. This effect was also dose-dependent [475].

The multi-kinase inhibitors sorafenib and lenvatinib are considered as first-line treatment. A combination of atezolizumab and bevacizumab has been recently proposed as a first-line treatment, but results are not impressive and many additional drugs have been tested. A recent meta-analysis suggested that regorafenib and cabozantinib may be the best candidates as second-line treatments in HCC [476].

However, there is extensive evidence that autophagy and ferroptosis are involved in the resistance of HCC to drugs, and their manipulation may improve the efficacy of treatments [477].

Autophagy inhibition may also be used as the treatment of HCC. GNS561, a new autophagy inhibitor, specifically inhibits the enzyme palmitoyl-protein thioesterase 1, (PPT1), leading to lysosomal membrane permeabilization, caspase activation and cell death [478].

Furthermore, the activation of the CD8+ T cells can induce ferroptosis by the suppression of the two components of the Xc- system [479]. RSL3, another ferroptosis inducer, also inhibits the proliferation of HCC cells [480]. Sorafenib resistance has been the most extensively investigated. However, investigations offered conflicting results as autophagy induced either increased resistance or increased efficacy in HCC sorafenib administration [481]. Sorafenib induces the ferroptosis of HCC cells due to the inhibition of the X−C system, followed by glutathione depletion. Ferroptosis inhibitors, such as ferrostatin-1, blocked the cellular death induced by sorafenib [482]. Sorafenib, combined with an aspirin treatment, synergistically induces apoptosis by blocking ACSL4 expression in HCC cells [456]. It was also found that the suppression of MT-1G leads to increased lipid peroxidation and sorafenib-induced ferroptosis in HCC cells [453]. Recent studies have shown the implication of the Yes-associated protein (YAP) in sorafenib resistance. The YAP/TAZ and ATF4 proteins are localized in the cytoplasm and antioxidant genes, such as SLC7A11, are not induced in sorafenib-sensitive cells and ferroptosis is increased. In sorafenib-resistant cells, however, YAP/TAZ and ATF4 are translocated to the nucleus and induce the SLC7A11 gene that represses ferroptosis [483]. Other factors associated with sorafenib resistance are hypercholesterolemia and the overexpression of the cholesterol sensor SCAP [484], and the high expression of the long non-coding RNA SNHG16 in association with low miR-23b-3p expression, leading to increased autophagy and apoptosis inhibition [485]. By contrast, the overexpression of miR-23a-3p directly targets ACSL4, leading to the suppression of ferroptosis and sorafenib resistance [486], while the dysregulation of miR-541 favors autophagy and increases sorafenib resistance [487]. FOXO3 upregulation increased autophagy and sorafenib resistance. Interestingly, the second-line drug regorafenib abolished this protective mechanism [488].

Recently, the variant 1 (tv1) of proliferating cell nuclear antigen clamp-associated factor (PCLAF) was found to reduce ferroptosis in HBV associated by decreasing Fe^2+^ accumulation [489]. On the other hand, the modulation of autophagy and/or ferroptosis may lead to an increased efficacy of sorafenib. Thus, quiescin sulfhydryl oxidase 1 increases ferroptosis and improves sorafenib efficacy [490].

Cholesterol reduces the degradation of the Golgi membrane protein 1 (GOLM1) and suppresses the GOLM1-dependent autophagy of receptor tyrosine kinases (RTKs), thus promoting HCC metastasis. Statins may, therefore, improve the efficacy of multiple tyrosine kinase inhibitors in HCC treatment [491].

CDGSH iron sulfur domain 2 (CISD2) is an iron-sulfur protein. The inhibition of CISD2 increased sorafenib-induced ferroptosis in resistant cells through either ferritinophagy or the inhibition of the p62–Keap1–NRF2 pathway [492]. The downregulation of complexin II (CPLX2) and haloperidol (a sigma receptor 1 antagonist) promotes the ferroptosis and cell death induced by sorafenib [493,494].

Autophagy inhibition improves sorafenib efficacy [495], while mitophagy induction increases sorafenib and lenvatinib resistance [431,496,497]. On the other hand, the downregulation of COX-2 by ketoconazole leads to mitophagy induction through the PINK–Parkin pathway and apoptosis stimulation [498].

Regorafenib resistance is due to reduction in the drug-induced apoptosis by topoisomerase IIα (TOP2A)-upregulated gene, which is involved in the resistance to regorafenib [499]. Interestingly, many natural products are effective, inhibiting protective autophagy or inducing autophagic death and the apoptosis of HCC cells [256].

Thus, heteronemin, a marine terpenoid, can induce ferroptosis in HCC cells [500].

The inhibition of the PI3K/AKT/mTOR pathway and the induction of autophagy and apoptosis is the mechanism of HCC anti-tumor effect of compounds, such as aloin (in combination with metformin), pueraria flavonoids, apigenin and Shikonin [354,360,501,502].

Solamargine has been shown to induce autophagy and apoptosis and inhibit HCC proliferation [309]. However, it should be noted that the stimulation of both apoptosis and autophagy may be detrimental, as autophagy supports HCC proliferation. Therefore, the combination with an autophagy inhibitor may be necessary as in the case of myricetin, which is a natural flavonoid [503].

This was not the case with sarmentosin, which induced caspase-mediated apoptosis in HCC cells blocked by the autophagy inhibitor chloroquine or the inhibition of Atg7, indicating that autophagy was important for sarmentosin efficacy. Mechanistically, sarmentosin inhibited mTOR and activated Nfr2 [504]. A detailed description of the mechanisms of drug resistance in HCC was recently published [505]. It should be stressed, however, that the above findings are based on experimental evidence and have not been tested in real life clinical trials.

## 8. Conclusions

Autophagy and apoptosis are two forms of regulated cell death. They are critically implicated in the regulation of HCC biology. Autophagy is interrelated with apoptosis and chemotherapy in HCC. Generally, the induction of autophagy inhibits caspase-dependent apoptosis, and the induction of apoptosis-associated caspase activation blocks the autophagic process. However, autophagy may also induce apoptosis. During HCC induction, autophagy acts as a tumor suppressor, but after induction, it behaves as a tumor promoter. Recently, ferroptosis, a separate form of regulated cell death, was identified. Despite its extensive implication in HCC, its interplay with autophagy and apoptosis, described in other conditions, has not been fully exploited in HCC. There are several switches that control the way in which the balance between autophagy and apoptosis turns. However, the initial cellular sensors that decide the direction of these two pathways have not yet been identified. A better clarification of the mechanisms involved may have clinical implications. The manipulation of either autophagy or apoptosis will improve the treatment outcomes of a difficult-to-treat tumor. There is a need to test, in clinical trials, substances that have been effective in experimental animals.

## Figures and Tables

**Figure 1 biomedicines-11-01166-f001:**
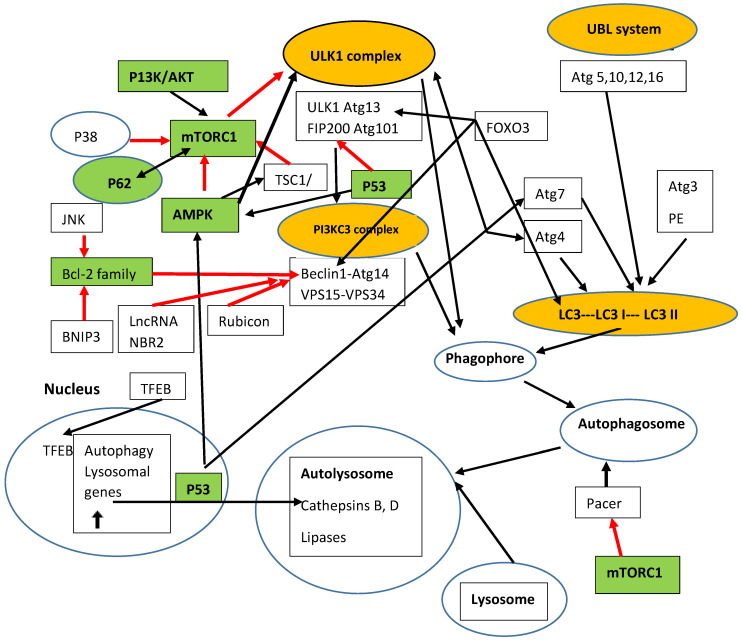
Regulatory pathways of autophagy. Black arrows indicate induction and red arrows indicate inhibition. See text for details.

**Figure 2 biomedicines-11-01166-f002:**
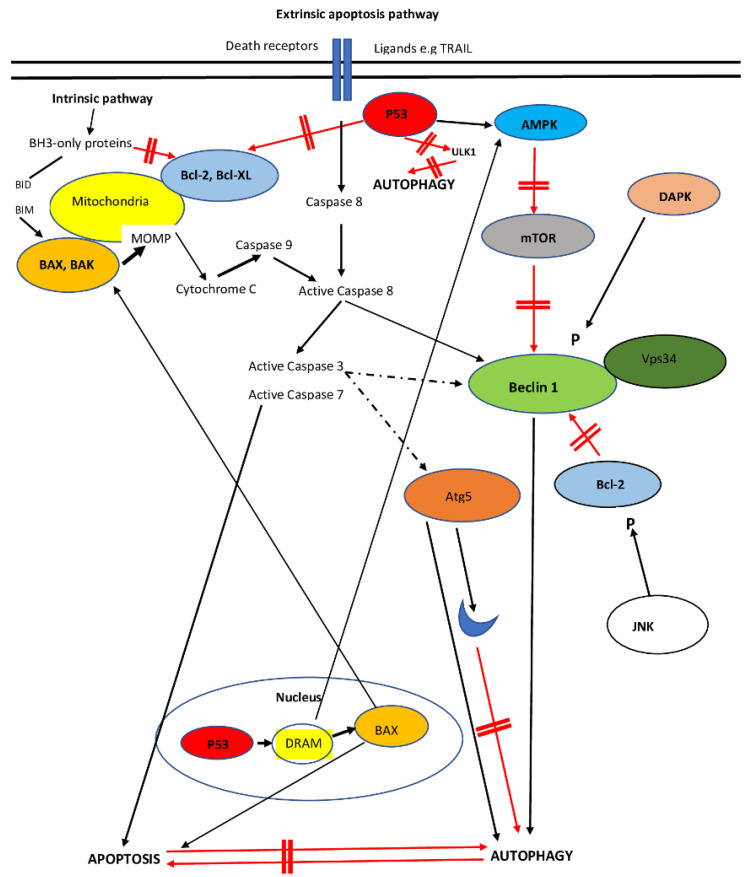
A simplified diagram of the interplay of autophagy and apoptosis. Most controllers of the interplay are presented. p53, the main gatekeeper, increases autophagy directly or indirectly through the DRAM activation of AMPK, but cytoplasmic p53 may inhibit autophagy. p53 increases apoptosis via the inhibition of Bcl-2 or the overexpression of BAX. Bcl-2, associated with Beclin1, inactivates the complex, leading to increased autophagy (only Vps34 is shown here). The phosphorylation of either Beclin1 by DAPK or Bcl-2 by JNK liberates the pro-autophagy complex. DAPK also increases apoptosis by an unknown mechanism. The cleavage of Beclin1 or Atg5 (an autophagy inducer promoted by ER stress) by apoptosis-induced caspases inhibits autophagy. Several intermediate components have been omitted for clarity. Intermittent line: Cleavage.

**Figure 3 biomedicines-11-01166-f003:**
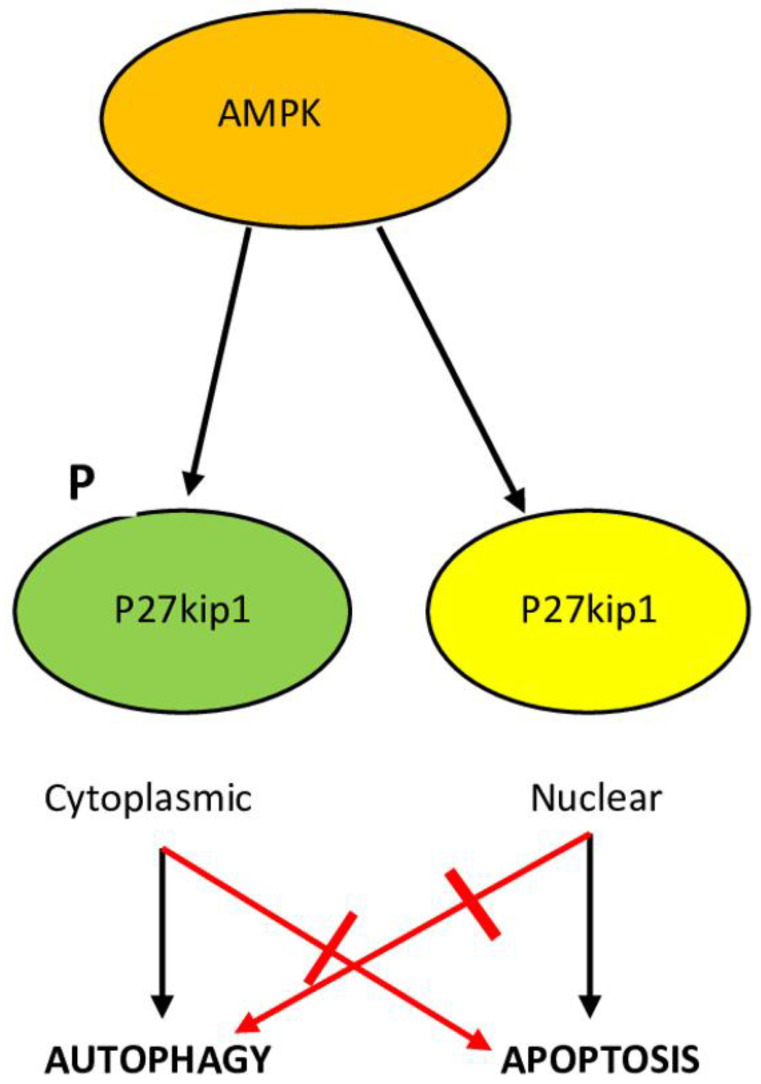
The p27 switch between autophagy and apoptosis is dependent on the phosphorylation and cellular localization of p27kip1.

**Figure 4 biomedicines-11-01166-f004:**
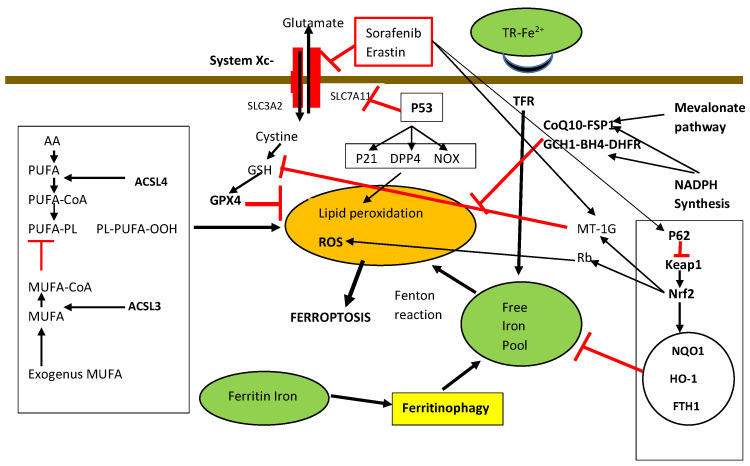
Mechanisms of ferroptosis. The Xc–antiporter system consisting of the SLC7A11 and SLC3A2 subunits allows for the extrusion and internalization of glutamate and cysteine. Glutathione peroxidase 4 (GPX 4) is produced by the glutamate–cystine exchange system. Xc- is the main inhibitor of ROS. See text for details.

## Data Availability

Not applicable.

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
