# Peer review of "Pathogenesis of Hepatocellular Carcinoma: The Interplay of Apoptosis and Autophagy"

_biomedicines, 2023, doi:10.3390/biomedicines11041166_

Round 1

Reviewer 1 Report

This review discusses the multifactorial pathogenesis of Hepatocellular Carcinoma (HCC) and the role of autophagy and apoptosis in regulating cell survival or death. The dysregulation of the balance between apoptosis and autophagy is common in HCC and can affect the fate of cancer cells. The review also highlights new developments, including the role of Endoplasmic Reticulum stress, microRNAs, and the gut microbiota. The article explores the characteristics of HCC associated with specific liver diseases and provides a brief description of autophagy and apoptosis. It reviews the experimental evidence indicating an interplay between autophagy and apoptosis and their role in the initiation, progression, and metastatic potential of HCC. Additionally, the review presents the role of ferroptosis and the potential therapeutic implications of autophagy and apoptosis in drug resistance.

  • While the review provides a concise overview of the pathogenesis of HCC and the role of autophagy and apoptosis, it may not cover all the current knowledge and developments in this field. Therefore, some readers may find the review lacking in detail or depth.
  • The article mentions the interplay between autophagy and apoptosis in HCC, but it does not explain the mechanisms of this interaction in sufficient detail. This may make it challenging for readers who are not familiar with these cellular pathways to fully comprehend their roles in HCC.
  • The review briefly mentions the role of ferroptosis in HCC, but it does not provide a comprehensive explanation of this regulated cell death mechanism. Therefore, readers may miss out on the potential importance of this pathway in HCC pathogenesis and treatment.
  • The article does not provide any state-of-the-art figures or diagrams to aid in visualizing the complex interactions between cellular pathways and their dysregulation in HCC. This may make it difficult for readers to understand the concepts and ideas presented in the text.
  • The introduction discusses hepatocellular carcinoma (HCC), which is a significant health problem worldwide with a high incidence rate and mortality rate. In 2020, there were over 905,677 new cases and 830,180 HCC-related deaths reported, and it is predicted that there will be more than 1 million deaths from HCC by 2030. HCC is often associated with liver diseases such as chronic viral hepatitis B and C, non-alcoholic steatohepatitis (NASH), and alcoholic liver disease (ALD). The risk factors for HCC vary depending on geographical location, with HCV, HBV, NAFLD/NASH, and ALD being the most common causes of HCC in the United States. In Asia, NAFLD-associated HCC is less common compared to the West.

    The development of HCC is associated with various forms of cellular death, including programmed cell death (PCD) such as apoptosis, necroptosis, and autophagy-dependent cell death, or accidental cell death such as pyroptosis and necrosis. Apoptosis is the most common form of PCD and does not elicit inflammation due to apoptotic bodies being engulfed by macrophages and degraded by lysosomes in autophagy. Autophagy is an important cellular degradation process that leads to the recirculation of structural components of the cell and improved survival. Autophagy-dependent cell death is a rare type of PCD, and lysosomes are the most important subcellular structure for the autophagic degradation of protein aggregates. Ferroptosis is a different type of PCD that depends on excessive iron and lipid peroxidation and is closely related to a specific form of autophagy called ferritinophagy, which participates in the turnover of cellular iron through the autophagic degradation of ferritin. Additionally, other forms of autophagy such as lipophagy and heat shock protein 90 (HSP90)-mediated chaperone-mediated autophagy (CMA) may induce ferroptosis by promoting lipid peroxidation. Thus, the introduction provides a comprehensive overview of the epidemiology and etiology of HCC, and the role of different forms of programmed cell death (PCD) in the development of HCC. However, the purpose of the article or research question is missing. It is also unclear how the various forms of PCD, including apoptosis, autophagy, pyroptosis, and ferroptosis, are related to HCC and how they may be targeted for therapeutic interventions. The article would benefit from a clear research question or objective and a more focused discussion of the role of PCD in HCC.

  • The pathogenic overview regarding autophagy can be improved by providing more specific details on the molecular mechanisms and signaling pathways involved in autophagy regulation. For instance, the text could discuss the role of specific autophagy-related genes and proteins such as ATG5, Beclin-1, and p62/SQSTM1 in the development of HCC. Additionally, the text could delve into the crosstalk between autophagy and other signaling pathways such as the PI3K/AKT/mTOR pathway and the p53 pathway. Furthermore, the text could discuss the role of autophagy in modulating the tumor microenvironment, such as its effect on immune cells and cytokine signaling. Finally, the text could provide a more comprehensive overview of the clinical implications of autophagy modulation in HCC, including potential therapeutic targets and the use of autophagy inhibitors or inducers in HCC treatment.

In summary, while the review presents an informative overview of the pathogenesis of HCC and the roles of autophagy and apoptosis, there may be limitations in the depth of the discussion and the lack of visual aids. Future revisions of the article could address these limitations by providing more detail and incorporating state-of-the-art figures and diagrams to enhance understanding.

Research in HCC field is correlated as it discusses the role of apoptosis and autophagy in regulating liver cell turnover and maintaining intracellular homeostasis in HCC, and also the treatment of advanced HCC with second-line therapies that have shown promising results in prolonging overall survival and improving progression-free survival. The dysregulation of apoptosis and autophagy is implicated in the development and progression of HCC, making it crucial to understand the treatment options available and their impact on cancer progression in managing HCC and preventing its metastasis (please refer to PMID: 34146196 and expand.

Author Response

Point by point response to Reviewer 1

Thank you very much for your constructive comments. All modifications of the text have been highlighted in yellow.

1.While the review provides a concise overview of the pathogenesis of HCC and the role of autophagy and apoptosis, it may not cover all the current knowledge and developments in this field.

Autophagy has been extensively revised and a new detailed figure has been added/revised.

2.The article mentions the interplay between autophagy and apoptosis in HCC, but it does not explain the mechanisms of this interaction in sufficient detail

Interplay details have been added including effects on HCC microenvironment.

3.The review briefly mentions the role of ferroptosis in HCC, but it does not provide a comprehensive explanation of this regulated cell death mechanism.

The section on ferroptosis has been extensively revised and all described mechanisms have been added in the revised figure.

  1. The article does not provide any state-of-the-art figures or diagrams to aid in visualizing the complex interactions between cellular pathways and their dysregulation in HCC.

Figs 1 and 4 contain detailed information after an extensive revision.

  1. It is also unclear how the various forms of PCD, including apoptosis, autophagy, pyroptosis, and ferroptosis, are related to HCC and how they may be targeted for therapeutic interventions. The article would benefit from a clear research question or objective and a more focused discussion of the role of PCD in HCC.

A clear objective has been added at the end of the introduction. An additional detailed subsection on how autophagy and apoptosis or their manipulation affect the response to treatment of HCC has been added.

6.The pathogenic overview regarding autophagy can be improved by providing more specific details on the molecular mechanisms and signaling pathways involved in autophagy regulation. For instance, the text could discuss the role of specific autophagy-related genes and proteins such as ATG5, Beclin-1, and p62/SQSTM1 in the development of HCC. Additionally, the text could delve into the crosstalk between autophagy and other signaling pathways such as the PI3K/AKT/mTOR pathway and the p53 pathway.

P62, p53 and the PI3K/Akt/mTor pathway have been additionally explained.

7.Finally, the text could provide a more comprehensive overview of the clinical implications of autophagy modulation in HCC, including potential therapeutic targets and the use of autophagy inhibitors or inducers in HCC treatment. The dysregulation of apoptosis and autophagy is implicated in the development and progression of HCC, making it crucial to understand the treatment options available and their impact on cancer progression in managing HCC and preventing its metastasis (please refer to PMID: 34146196 and expand.

It has been done. Please refer to the response at no.5

Reviewer 2 Report

Very interesting paper and comprehensive review. My comments:

1) English grammar should be improved. I recommend the authors to have their manuscript revised by a native speaker

2) I recommend to add more comments on specific disease-related HCC, for example in patients with hemocromatosis .....

3) The authors should comment also the role of non-liver related diseases on HCC occurrence, for example diabetes.

4) The authors should also comment on the role of some drugs on HCC occurrence (for example statins, citPMID: 32260179)

5) THe authors should comment on how epidemiology of HCC is changing based on these findings, in favor of NAFLD-related HCC (cite the recent nationwide paper PMID: 33219585)

Author Response

Point by point response to Reviewer 2

Thank you very much for your constructive comments. All modifications of the text have been highlighted in yellow.

  • English grammar should be improved. I recommend the authors to have their manuscript revised by a native speaker

The manuscript has been re-checked.

  • I recommend to add more comments on specific disease-related HCC, for example in patients with hemochromatosis.

 Hemochromatosis has been added to the text.

  • The authors should comment also the role of non-liver related diseases on HCC occurrence, for example diabetes.

Diabetes and HCC have been added to the text.

  • The authors should also comment on the role of some drugs on HCC occurrence (for example statins, citation PMID: 32260179)

Statins as possible preventive drug for HCC have been commented on.

  • The authors should comment on how epidemiology of HCC is changing based on these findings, in favor of NAFLD-related HCC (cite the recent nationwide paper PMID: 33219585)

A paragraph on the changing epidemiology of HCC has been added.

Round 2

Reviewer 1 Report

I am satisfied with the rebuttal provided.

Reviewer 2 Report

The revised manuscript is OK. Thank you!